# Insights into the Oxidative Stress Response of *Salmonella enterica* serovar Enteritidis Revealed by the Next Generation Sequencing Approach

**DOI:** 10.3390/antiox9090849

**Published:** 2020-09-10

**Authors:** Xiaoying Liu, Misara Omar, Juan E. Abrahante, Kakambi V. Nagaraja, Sinisa Vidovic

**Affiliations:** 1Department of Veterinary and Biomedical Sciences, University of Minnesota, Saint Paul, MN 55108, USA; liux2725@umn.edu (X.L.); omarx119@umn.edu (M.O.); nagar001@umn.edu (K.V.N.); 2University of Minnesota Informatics Institute, University of Minnesota, Minneapolis, MN 55455, USA; abrah023@umn.edu; 3The New Zealand Institute for Plant and Food Research, 1025 Auckland, New Zealand

**Keywords:** non-typhoidal *Salmonella*, oxidative stress response, next generation sequencing, biofilm formation, salmonellae-essential virulence genes

## Abstract

As a facultative intracellular pathogen, *Salmonella* Enteritidis must develop an effective oxidative stress response to survive exposure to reactive oxygen species within the host. To study this defense mechanism, we carried out a series of oxidative stress assays in parallel with a comparative transcriptome analyses using a next generation sequencing approach. It was shown that the expression of 45% of the genome was significantly altered upon exposure to H_2_O_2_. Quantitatively the most significant (≥100 fold) gene expression alterations were observed among genes encoding the sulfur utilization factor of Fe-S cluster formation and iron homeostasis. Our data point out the multifaceted nature of the oxidative stress response. It includes not only numerous mechanisms of DNA and protein repair and redox homeostasis, but also the key genes associated with osmotic stress, multidrug efflux, stringent stress, decrease influx of small molecules, manganese and phosphate starvation stress responses. Importantly, this study revealed that oxidatively stressed *S.* Enteritidis cells simultaneously repressed key motility encoding genes and induced a wide range of adhesin- and salmonellae-essential virulence-encoding genes, that are critical for the biofilm formation and intracellular survival, respectively. This finding indicates a potential intrinsic link between oxidative stress and pathogenicity in non-typhoidal *Salmonella* that needs to be empirically evaluated.

## 1. Introduction

Non-typhoidal *Salmonella* is a leading cause of foodborne gastroenteritis on the global scale. On the global scale, this foodborne pathogen is responsible for 80 million cases of gastroenteritis annually [1]. The situation in the USA is very similar to the global epidemiological picture of salmonellosis caused by non-typhoidal *Salmonella*. It has been estimated that from 2000 to 2008, non-typhoidal *Salmonella* serovars accounted for 1.2 million laboratory-confirmed illnesses, 19,000 hospitalizations, and 380 deaths each year in the USA [2]. The high incidence rate is made more significant by the fact that there have been no signs of decline in the incidence of salmonellosis over the past 15 years in the US. Intriguingly, infections caused by other five major food-borne pathogens (i.e., *Escherichia coli* O157:H7, *Campylobacter* spp., *Listeria monocytogenes*, *Yersinia enterocolitica*, and *Shigella* spp.) declined substantially over the same time [2].

Infections of humans with non-typhoidal *Salmonella* serovars occur mostly via the human food chain [3,4]. The important human food chain transmission route of non-typhoidal *Salmonella* is associated with ready-to-eat products such as vegetables [5,6], leafy green horticulture products [7,8], and fruits [9,10]. Traditionally, aqueous chlorine and ozone solutions are widely used in the food industry for sanitization of food-processing surfaces as well as vegetables and fruits prior to their distribution to the market [11,12]. Chlorine and ozone, potent oxidizing agents, impose an oxidative stress on cells, damaging nucleic acids, both DNA and RNA, proteins, lipids, and further causing membrane disorganization and genome breaks which result in cellular death. To survive oxidative stress assault in the food matrix and food processing facilities and successfully invade the host and survive intracellularly, these pathogens must acquire a robust oxidative stress protective mechanism.

A number of studies investigated the roles of transcription factors [13], RNA repair systems [14], sigma factors [15], and novel stress response proteins [16,17] in the oxidative stress response of non-typhoidal *Salmonella*. Although these studies gave us extremely valuable information on the functions of specific oxidative response mechanisms employed by these pathogens, they could not provide insights into the global impact of oxidative stress on the non-typhoidal *Salmonella* physiology. Recently, Fu et al. [18] by employing global comparative proteomics, identified 116 proteins in the proteome of *Salmonella* Typhimurium, treated with H_2_O_2_ that were significantly altered compared to the control. Among the altered proteome, iron acquisition systems were most significantly induced. Also, several phage-coding proteins with DNA repair capabilities were induced, whereas the apparatus of the *Salmonella* type III secretion system was repressed.

To gain insight into the expression dynamics of genes associated with distinct biological functions during an oxidative stress, expression of ten genes, including *iroN*, *sitA*, *rpoS*, *rpoH*, *ycfR*, *dps*, *nrdM*, *pocR*, and *ompF*, were tested after 30 min of 1 mM, 2 mM, and 4 mM H_2_O_2_ treatments. The 30 min of H_2_O_2_ treatment was selected as *Salmonella enterica* subspecies *enterica* serovar Enteritidis, hereinafter *Salmonella* Enteritidis cultures were already adapted at that point to different H_2_O_2_ treatments, showing only a slight decline in the viability. Finally, to determine the global impact of oxidative stress on the non-typhoidal *Salmonella* physiology, we carried out comparative transcriptome analyses using the next generation sequencing approach.

## 2. Materials and Methods

### 2.1. Bacterial Culture

*Salmonella* Enteritidis ATCC 13076 (American Type Culture Collection, Manassas, VA, USA) was used as the model organism. For each experiment, fresh culture was prepared by streaking frozen culture onto tryptic soy agar (EMD Chemicals Inc., Darmstadt, Germany), followed by incubation at 37 °C for 24 h. Three well-isolated colonies were used for subsequent inoculations.

### 2.2. Oxidative Stress Killing Assay

This assay was carried out using overnight cultures of the wild type *S*. Enteritidis strain grown in LB at 37 °C with continuous shaking at 180 rpm. These overnight cultures were diluted 1/100 in LB followed by growth to 0.4 using an optical density at 600 nm. The wild type cultures were exposed at their mid-exponential growth phase (0.4) to three incrementally increasing concentrations of H_2_O_2_, including: 1 mM, 2 mM, and 4 mM. Treatment was carried out for 90 min under the same incubation conditions. Wild type culture with no oxidative treatment was included as a control. Colony forming units (CFU) counts were carried out by 10-fold dilutions using sterile saline at time zero (before H_2_O_2_ exposure) and then at every 15 min of treatment. Aliquots of 0.1 mL were plated in triplicate followed by incubation at 37 °C for 24 h.

### 2.3. Incremental Oxidative Stress Gene Expression Assay

Cultures of *S*. Enteritidis were prepared as described above and after 30 min of oxidative treatment samples were taken and processed. Total RNA was isolated as described below followed by cDNA synthesis using Superscript™ III reverse transcriptase (RT) (Invitrogen by Life Technologies, Carlsbad, CA, USA). Quantitative PCR was carried out on a MiniOpticon^TM^ Real-Time PCR Detection System (Bio-Rad Laboratories, Hercules, CA, USA) with iQTM SYBR Green Supermix kit (Bio-Rad Laboratories) as described previously [19]. Genes *rtcR* and *yjjA*, encoding for transcriptional regulatory protein RtcR and DUF2501 domain-containing protein, were selected as the internal reference genes to normalize the expression of the tested genes. Primers were designed to target expression of the following genes: *rpoS*, *rpoH*, *iroN*, *sitA*, *dps*, *nrdM*, *trxC*, *ycfR*, *ompF*, and *pocR* (Table 1). These ten genes were selected based on their distinct gene ontologies (GO) associated with the oxidative stress response. Selected the GO include, iron homeostasis, alternative sigma factors, multiple stress response, DNA protection, antioxidants, DNA replication, uptake of small solutes and anabolic processes. Melting curves were analyzed to check the product specificity. The level of gene expression for each target gene was compared with the internal reference control gene and the fold changes in expression were calculated by the comparative *CT* method [20].

### 2.4. Preparation of Samples for Transcriptomics

Untreated and 30-min H_2_O_2_ (3 mM) treated cultures (0.4 at OD_600_) of *S*. Typhimurium were centrifuged at 4000 rpm for 5 min. Supernatants were discard and the cell pellets were washed two times followed by RNA extraction using the RNeasy Mini kit (Qiagen Inc., Valencia, CA, USA) according to the manufacturer’s instructions.

### 2.5. Global Transcriptomic Analysis

Sample quality preparation was done using capillary electrophoresis (Agilent Bio Analyzer 2100, Santa Clara, CA, USA) as previously described [17]. Only samples that had RNA integrity number 8 or greater were considered for further analysis. Illumina sequencing libraries were prepared using Illumina’s TruSeq Stranded Total RNA Library Prep Human/Mouse/Rat Sample Preparation Kit (Cat. # 20020597) at the University of Minnesota Genomics Center (Saint Paul, MN, USA). For rRNA depletion, 500 ng of total amount of RNA was used in combination with Ribozero capture probes. After this step, the mRNA was fragmented followed by reverse transcription into complementary DNA (cDNA). The resulting cDNA fragments were coded with indexed adaptors followed by amplification using 15 PCR cycles. Again capillary electrophoresis was used to validate library size distribution, while the library quantification was carried out using fluorimetry. Libraries normalization was carried out followed by their hybridization to a single read flow cell. Once hybridization completed, the flow cell was loaded on the HiSeq 2500 and sequenced using Illumina’s SBS chemistry. The primary analysis and index de-multiplexing were performed as previously described [17]. Read mapping was carried out using publicly available genome of *S*. Enteritidis P125109 strain. Quantification of gene expression was completed using Feature Counts. Significantly differentially expressed genes were identified using the edgeR feature in CLCGWB (Qiagen, Valencia, CA, USA) based on a minimum 2 times absolute fold change difference and *p* < 0.05 false discovery rate (FDR).

### 2.6. Validation of RNA-Seq Data by Real-Time PCR

Cultures of *S*. Enteritidis were prepared as described above and after 30 min of oxidative treatment samples were taken and processed. Total RNA was isolated as described above. Quantitative PCR assay was carried out as described above. Genes *rtcR* and *yjjA* were selected as the internal reference genes to normalize the expression of the tested genes. Primers were designed to target expression of the following genes: *dinP*, *dnaJ*, *eutC*, *mdlA*, *osmY*, *pagO*, *spaR*, *malK*, and *nrdD* (Table 1).

### 2.7. Experimental Replication and Gene Ontology Analysis

The data of all experiments represent the average of three biological replications. The cell viability during the oxidative treatments were analyzed by CoStat version 6.4 software (Co-Hort Software, Monterey, CA, USA) using the homogeneity of linear regression slopes approach. The gene ontology (GO) analysis was carried out using the Database for Annotation, Visualization and Integrated Discovery (DAVID) database [21] as well as the National Center for Biotechnology Information (NCBI).

### 2.8. RN-Seq Accession Numbers

RNA sequencing data for all three biological replications and two different treatments were deposited in the NCBI under accession number GSE 155479.

## 3. Results

### 3.1. Effect ofIincreasing H_2_O_2_ Concentrations on the Viability of S. *Enteritidis*

To evaluate the ability of *S*. Enteritidis to survive exposure to oxidative stress, we performed oxidative stress killing assays with increasing concentrations of H_2_O_2_ over a short time. The concentration of 1 mM of H_2_O_2_ caused a death rate of 0.37 log_10_ CFU for the first 15 min of exposure (Figure 1 and Appendix A). After this initial lethal period, the culture of *S*. Enteritidis exposed to the same concentration of H_2_O_2_ showed a slight decline in the number of viable cells, resulting in only 0.1 log_10_ CFU of reduction during the entire experiment (Figure 1). The exposure of *S.* Enteritidis to 1 mM of H_2_O_2_ resulted in two distinct survival phases, an initial and brief lethal phase followed by a long bacteriostatic phase (Figure 1 and Appendix A). Similarly to 1 mM, the concentration of 2 mM of H_2_O_2_ caused an initial lethal phase resulting in 0.53 log_10_ CFU decline (Figure 1). However, the survival pattern associated with the 2 mM H_2_O_2_ treatment was bactericidal not bacteriostatic (Figure 1). After an initial lethal period for the first 15 min of H_2_O_2_ exposure, the 2 mM H_2_O_2_ treatment continued to cause cell death, albeit at a slower rate compared to that of the initial 15-min period. This bactericidal phase was characterized by repetitive fluctuations of fast die-off periods (e.g., 0.23, 0.24 and 0.22 log_10_ CFU declines at the 30-min, 60-min, and 90-min measurements, respectively) and slight growth periods (e.g., 0.06 log_10_ CFU increase at the 45-min measurement and 0.03 log_10_ CFU increase at the 75-min measurement) (Figure 1). The exposure of *S*. Enteritidis to the greatest concentration of H_2_O_2_, 4 mM, caused again an initial lethal phase, characterized by a fast die-off rate, 0.72 log_10_ CFU decline over the first 15 min of treatment (Figure 1). Interestingly, this initial fast die-off period was followed by moderate die-off periods (e.g., 0.18 and 0.17 log_10_ CFU decline) at the next two measurements (Figure 1). After this moderate die-off period, the culture of *S*. Enteritidis entered progressively fast mortality rates resulting in a 0.38 log_10_ CFU decline at the 60-min measurement and 1.19 log_10_ CFU decline at the 75-min measurement (Figure 1). At the last, 90-min, measurement, the viable cell count could not be detected at 10^3^ CFU per 1 mL, indicating a continuation of this progressive and fast mortality rate. The oxidative treatments with 1 mM, 2 mM, and 4 mM of H_2_O_2_ caused significantly higher (*p* < 0.01) mortality rates during the exponential growth phase of *S*. Enteritidis compared with no treatment of this pathogen under the same growth conditions. The average mortality rates of *S*. Enteritidis for every 15 min of H_2_O_2_ treatment were 0.08 log_10_ CFU decline for 1 mM, followed by 0.188 log_10_ CFU decline for 2 mM treatment and 0.468 log_10_ CFU decline for 4 mM. Although all three H_2_O_2_ treatments caused significantly higher mortality rates compared to that of no treatment, the survival patterns of H_2_O_2_ treated *S*. Enteritidis were different (Figure 1).

### 3.2. Expression Levels of the Genes Associated with Oxidative Stress and Anabolic Processes during Incremental Increase of H_2_O_2_ Concentration

To determine the relationship between a death rate of *S*. Enteritidis and expression of genes that are associated with the most important oxidative stress adaptive processes, we carried out a qRT-PCR assay. The gene expression assay was carried out after 30 min of H_2_O_2_ treatment when *Salmonella* cells have already been adapted to the oxidative stressor (Figure 1). The targeted genes were selected from the following stress response and anabolism associated processes, iron homeostasis (*iroN*, *sitA*), alternative sigma factors (*rpoS*, *rpoH*), multiple stress responses (*ycfR*), DNA protection (*dps*), antioxidants (*trxC*), DNA replication (*nrdM*), uptake of small solutes (*ompF*) and anabolic process (*pocR*).

No significant changes in expression of *rpoS* were observed during 1 mM and 2 mM H_2_O_2_ treatment, whereas 4 mM H_2_O_2_ treatment caused a significant (2.5-fold) downregulation of the *rpoS* gene (Figure 2). Similarly to the *rpoS* gene, the oxidative treatments of *S*. Enteritidis caused minor alterations in the expression of the gene that encodes another stress response sigma factor, RpoH (Figure 2). The 1 mM and 4 mM treatments caused 2-fold and 2.9-fold upregulation of the *rpoH* gene, while 2 mM treatment caused no significant (1.4-fold) upregulation of the same gene.

In sharp contrast to the genes that encode stress response sigma factors, the genes associated with iron homeostasis, in particular genes that encode the iron receptors, IroN (out membrane receptor) and SitA (periplasmic receptor) showed the greatest gene expression alterations (Figure 2). Most notably, the expression of *iroN* exhibited a profound upregulation with the 1 mM H_2_O_2_ treatment (532-fold) and with 2 mM H_2_O_2_ treatment, the upregulation of this gene was extraordinarily high (75664-fold) compared to that of the H_2_O_2_ untreated control. Interestingly, with the greatest concentration of oxidative stressor, 4 mM, the upregulation of the *iroN* gene was lower (292-fold) than that with 1 mM (532 fold) and 2 mM (75664 fold) treatment (Figure 2). Similarly, other genes involved in the iron homeostasis, *sitA*, showed large upregulation upon the oxidative stress treatment with 1 mM (298-fold), 2mM (316-fold) and 4 mM (675-fold) H_2_O_2_ treatment. Although both genes encode for iron receptors, their expression (stress response) patterns were different. The upregulation of the *sitA* gene was proportional to incremental increases of H_2_O_2_ concentration, whereas the upregulation of the *iroN* gene did not follow the level of H_2_O_2_ concentration (Figure 2).

The *ycfR* gene, which encodes a multiple stress response protein YcfR, was upregulated throughout the H_2_O_2_ treatments. The upregulation of the *ycfR* gene ranged from 283-fold and 273-fold to 103-fold in the *Salmonella* cultures exposed to 1 mM, 2 mM, and 4 mM H_2_O_2_, respectively (Figure 2). Similarly to the *iroN* gene, the greatest concentration of H_2_O_2_, 4 mM, caused the lowest upregulation of the *ycfR* gene, further indicating to some extent the similar patterns of the responses of these two oxidative stress response associated genes. However, the *ycfR* gene did not show an extremely high upregulation during the 2 mM H_2_O_2_ treatment, as the *iroN* gene showed (Figure 2). Another stress response gene, *dps*, had an unusual expression profile. During treatment with 1 mM H_2_O_2_ this gene was upregulated 44-fold, then during the 2 mM treatment the *dps* gene was downregulated 4-fold, and again upregulated 8.9-fold with 4 mM H_2_O_2_ treatment (Figure 2). This unusual expression profile of the *dps* gene, which encodes a protein that protects DNA from reactive oxygen species (ROS), indicates a complex interaction of the oxidative stress response associated genes.

The genes associated with DNA replication and oxidoreductase processes, *nrdM* and *trxC*, had similar genes expressions profiles (Figure 2). Both genes exhibited 35- to 70-fold upregulation during the oxidative treatments (Figure 2). The *ompF* gene, which encodes for a major nonspecific OmpF porin, and the *pocR* gene, which is associated with anabolic processes, exhibited downregulation during the oxidative treatments (Figure 2). Interestingly, the downregulation gene expression patterns for both the *ompF* and *pocR* genes were the same. The greatest gene alterations (e.g., downregulation) were observed during 2 mM H_2_O_2_ treatment, whereas the 4 mM H_2_O_2_ treatment caused the lowest level of the gene expression alteration for both genes (Figure 2).

### 3.3. Global Transcriptome Response of S. *Enteritidis* to Oxidative Stress.

After the determination of the stress response patterns of a limited number of genes during treatment with different concentrations of H_2_O_2_, we carried out the global transcriptome analyses using a single concentration of H_2_O_2_. In total, 2051 genes were significantly differently expressed (SDE) in *S*. Enteritidis during the 3 mM H_2_O_2_ treatment compared to the reference transcriptome of the same *S*. Enteritidis strain with no H_2_O_2_ treatment. Of the 2051 SDE genes, 1111 were induced and 941 genes were repressed, showing at least a 2-fold alteration in gene expression and high reproducibility (false discovery rate < 0.5) across all three biological replications (Appendix A).

#### 3.3.1. Induction of Transcriptome—Molecular Response of *S*. Enteritidis to Oxidative Stress

The most altered change in biological processes of oxidatively stressed *S*. Enteritidis cells was the induction of genes encoding Fe-S cluster biogenesis (Appendix A). Operons encoding sulfur utilization factor (SUF), *sufABCSD* and *ynhAG*, were upregulated from 173- to 903-fold and from 23- to 114-fold, respectively (Figure 3). The SUF operons were profoundly more induced compared to ones encoding iron-sulfur cluster (ISC), *nifJUS*, *yfhFP* and *hscBA* (Figure 3). Another group of highly induced genes were associated with the iron ion homeostasis. The operon *sitABCD*, iron/manganese ABC transporter, showed the greatest induction among operons/genes associated with this biological function (Figure 3). Besides this Fe/Mn transporter, a group of operons/genes encoding a wide range of iron transporters showed the greatest inductions, including *iroN*, ferric salmochelin siderophore receptor; *fepBD*, ferric enterobactin ABC transporter; *fhuABCD*, ferrichrome outer membrane transporter; *ydiE*, hemin transporter; *exbBD*, iron siderophore bacteriocin transporter; *fhuE*, ferric coprogen and ferric-rhodotorulic acid receptor as well as *fhuF*, receptor for ferrioxamine B (Figure 3). Also, several operons and a gene responsible for the biosynthesis of salmochelin (*iroBCE*), enterobactin (*ybdABZ*, *entABCDEF*) and hem (*hemH*) were greatly induced during oxidative stress (Figure 3).

The significant induction showed a group of operons/genes encoding proteins that govern the cell-redox homeostasis. Among the cell-redox encoding genes, the *nrdH* gene which encodes a classical antioxidant, glutaredoxin-like protein NrdH, showed the greatest induction, resulting in 144.3-fold upregulation compared to the reference strain (Figure 4). The *trxC* gene that encodes an effective cytoplasmic disulfide-reducing protein, thioredoxin 2, showed 76-fold induction (Figure 4). The *grxAB* operon, encoding the antioxidants glutaredoxin 1 and 2, showed a moderate induction compared to that of the *nrdH* gene. A major part of the cell-redox regulon was associated with the genes encoding proteins associated with the electron transport chain (Figure 4). The *narIJHKG* operon, involved in nitrate assimilation and anaerobic electron transport, exhibited the greatest induction among this group of genes, followed by *fdoHGI*, dehydrogenases that transfer electrons from the periplasmic to the cytoplasmic heme; *sdhABCD*, cytochrome b556; *cyoABCDE*, cytochrome bo3 oxidase complex and *sthA*, which converts NADPH to NADH (Figure 4).

The most numerous group of altered operons/genes in the transcriptome of oxidatively stressed *S*. Enteritidis cells was associated with various stress response processes as a distinct biological function indicated by the GO analysis. The greatest alteration in the entire transcriptome of *S*. Enteritidis under H_2_O_2_ stress was related to the *ycfR* gene. This gene, encoding the multiple stress resistance outer membrane protein YcfR, was induced 913-fold compared to that of the reference strain (Figure 4). The great induction exhibited the *dps* gene, which encodes the DNA-binding protein Dps that protects DNA. Besides *dps*, another group of genes encoding DNA repair proteins was significantly induced, including the DNA damage inducible *yebG* gene and the *recAN*, *polB*, *dinPIG*, and *deoBCD* genes (Figure 4). Similarly to genes encoding DNA repair proteins, a group of genes encoding molecular chaperones and proteases was significantly induced. Among this group of genes, the *ibpBA* gene, encoding small heat shock proteins, showed the greatest induction, followed by molecular chaperones *dnaKJ, yhhW*, and *clpAB*, and protease *pgtE* (Figure 4).

The oxidative stress response regulon included upregulation of: Osmotically inducible genes *envFR*, *osmYC*, and *yjbJ*; efflux encoding genes *yheR*, *yabL*, *yhcQP, dinF*, and *mdlAB*; phosphate starvation inducible genes *phoHE, yjbA*, and *phnTSUR*; sensing and removal of ROS *katG*, *ahpCF, yhaK*, *sodAC, soxSR, yqhACDE*, and *yhcN*; adaptation to low Mg^2+^ environment *pagCDOK* and *mgtBC;* and the programmed cell death inducible gene *ybaJ* (Figure 4).

The oxidative stress caused the induction of numerous operons/genes located on *Salmonella* pathogenicity islands (SPIs). The significant induction was observed for the *ssrAB* operon that positively controls the expression of the SPI-2 genes (Figure 5) 

Indeed, a significant induction of the *ssaMCBDEVJKHILQN* operon encoding the type III secretion system (T3SS) apparatus of the SPI-2 as well as operons *sifAB* and *sseAbJBbFIE* encoding the SPI-2 effector proteins (Figure 5) was observed. Also, *spaQSROP* (translocon), *iagB*, *sinHR*, and *sopE2* (effectors) as well as *sicP* (effector chaperone) located on the SPI-1 were significantly induced (Figure 5). Not only genes encoding the T3SS apparatus and effector proteins responsible for the invasion of host cells and intracellular survival, but also genes encoding various cell adhesions were induced. This group of genes included *yncC*, biofilm transcriptional regulator; *stbABCD*, *fimHYWDF*, *sthAB* and SEN4247, fimbrial operons/genes; *safBCD* and SEN1978, pilin operon/gene as well as *csgABF*, curli operon (Figure 5).

Adaptation to oxidative stress involved induction of operons/genes that encode proteins associated with carbohydrate metabolism *yjfR*, *ydiN*, *nagABCE* (glucosamine catabolic processes), *pduJKLMDQTNOPHUWSECVXG* (carboxysomes formation) and *yiaMOGN* (uptake of gluconate); amino acid metabolism *asnABC*, *yehEWX*, *dadAX*, *yejF*, *lysA*, *yneHIG*, *ybaO*, and *livCKGFMH* (uptake of branched-chain amino acids); nucleoside metabolism *xapAB* (purine salvage pathway); outer membrane modification *pagP*, *nlpD*, *ybaN*, *wcaEFGHID* (colonic acid biosynthesis), *safA*, *ompX* (porin), and *ompA* (porin) and genes encoding proteins of unknown functions (Appendix A).

#### 3.3.2. Repression of Transcriptome—Molecular Response of *S*. Enteritidis to Oxidative Stress

There is a distinct segregation based on the biological functions between the induced and repressed transcriptome of oxidatively stressed *S*. Enteritidis cells. While the induced transcriptome showed a great association with the iron-sulfur cluster assembly, iron ion homeostasis, cell redox homeostasis, a variety of stress responses, pathogenesis and sessile lifestyle (cell adhesion), the repressed transcriptome was mainly associated with carbohydrate metabolic processes, ribosomal biogenesis, anaerobic respiration and cell motility (Figure 6).

The *tdcA* gene, encoding a transcriptional activator for the *tdcABCD* operon, showed the greatest repression, resulting in 131-fold downregulation compared to that of the reference strain (Figure 7). Indeed, other genes of the *tdcABCD* operon, which are involved in L-threonine and L-serine catabolic processes, showed significant repression (Figure 7). Also, additional operons/genes associated with amino acid metabolism showed significant repression including *yhdG* (amino acid permease), *gcvTHR* (glycine, serine and threonine metabolism), *pepETQD* (hydrolases of peptide bonds), *selAB* (selenocysteine synthesis) and *sdaBC* (serine uptake and degradation) (Figure 7).

Carbohydrate metabolism was the biological function most commonly affected within the repressed transcriptome. This group of operons/genes, encoding a wide range of proteins, was involved in catabolic pathways, including *gudDP* (D-glucarate), *garDLKR* (galactarate), *eutSPQT* (ethanolamine), *nanAEKT* (sialic acid), *melABR* (melibiose), *srlAR* (sorbitol), *kdgTK* (D-glucuronate), *idnDT*, and *gntRT* (D-gluconate) (Figure 7). Besides this group of operons/genes associated with catabolism, there was repression of operons/genes encoding sugar uptake and anabolism such as *malFTKMG* (maltose), *ydeWZYV* (carbohydrate permease), *yadBEI* (a major carbohydrate active transport system), *mglABC* (galactose/methyl galactoside) and *citCDEFD2E2* (citric operon involved in anabolic processes of the tricarboxylic acid cycle) (Figure 7).

Other central metabolic processes repressed during oxidative stress of *S. enterica* included anaerobic respiration (*napABCDFGH*, *dmsA2A3A1AB1B*, *glpABCFTXK*, *frdABC*, *nrfABCD*, *hybABCFG*, *deuABC*), ribosomal biogenesis and rRNA processing (*ygjRO*, *rpsUTPLGSOCFHNJQBR*, *rplUKSJNXVBCDAEPLWYM*, *rimJMK*, *rpmAGBHC*), fatty acid metabolic processes (*accBD*, *fadHRL*, *fabIGBAH*), vitamin biosynthesis (*pocR*, *ynfKA*, *cbiACDETF*, *menDAB*, *yfbBSU*), ATP synthesis (*atpBIEHFGA*), cellular protein modification and transport processes (*hypAODBCE* and *oppCBD*), aerobic respiration (*yhbUTCSPQV*), the ethanolamine catabolic process (*eutSPQT*), the carnitine metabolic process (*caiFEA*) and the pyrimidine nucleobase biosynthetic process (*pyrHDLE*) (Figure 6).

### 3.4. Validation of RNA-seq Data by Real-Time PCR

To validate the RNA-seq data, the expressions of nine randomly chosen genes were tested by quantitative real-time PCR (qRT-PCR) under the same experimental conditions. The qRT-PCR showed the same pattern of expressions, including seven upregulated genes (*dinP*, *dnaJ*, *eutC*, *mdlA*, *osmY*, *pagO*, and *spaR*) and two downregulated genes (*malK* and *nrdD*) (Figure 8). It can be observed that generally, the expression values measured by qRT-PCR were greater compared to those measured by RNA-seq analysis. The difference in expression values between these two techniques is consistent with the fact that qRT-PCR generally gives higher expression values compared to RNA-seq [17,22].

## 4. Discussion

*Salmonella* Enteritidis, one of the most important non-typhoidal *Salmonella* serovars from the public health and veterinary medicine perspectives [23,24,25,26], has developed a robust anti-oxidative response not only to survive endogenous oxidative stress [27] but also to survive exogenous stress imposed by a host [28] or environment [29]. In other words, for successful host invasion, intracellular proliferation and/or environmental survival, possession of an efficient oxidative stress response is one of the key physiological characteristics of this pathogen. In this study, it was shown that incrementally increased concentrations of oxidant agent caused various patterns of gene expressions. While inductions of genes encoding antioxidant (thioredoxin 2) and DNA replication regulator (NrdM) were constant regardless of H_2_O_2_ concentrations, induction of a gene encoding iron/manganese transporter, SitA, was proportional with H_2_O_2_ concentrations. In contrast to the pattern of *sitA* induction, the *ycfR* gene, encoding multiple stress-response protein, exhibited disproportionate induction regarding H_2_O_2_ concentration. The greatest concentration of H_2_O_2_ resulted in the lowest *ycfR* induction. Besides these three patterns of gene expressions, the most common gene expression alteration during this assay was associated with an unpredictable pattern, where the middle concentration of H_2_O_2_ caused the greatest gene expression alterations, either induction (*iroN*) or repression (*pocR*, *ompF* and *dps*). Recently, Mitosch et al. [30], using a dual-reported model, demonstrated the importance of timing of the gene expression during exposure of the microbial cell to an oxidative agent. Besides the importance of temporal gene expression response, this study points to a multifaceted effect of oxidative agent concentration on gene expression during oxidative stress, further indicating another dimension of cellular adaptation complexity to oxidative stress.

The comparative transcriptomic analyses revealed that the expression of 45% of the *S*. Enteritidis genome was altered at least two-fold upon exposure to H_2_O_2_. This finding showed that oxidative stress has a much greater impact on the physiology of this pathogen than previously anticipated [18,31,32]. Out of numerous biological functions altered by oxidative stress, the SUF mechanism of Fe-S cluster formation exhibited the greatest change. The Fe-S clusters are cofactors of proteins and they are involved in various critical metabolic processes including DNA replication, regulation, and repair, substrate uptake, respiration, and RNA modification [33]. In general, there are two Fe-S cluster formation systems, the ISC system and the SUF system [33]. In this study, the SUF system showed a significantly greater induction than the ISC system. This finding correlates with previous studies that pointed out that the ISC is a housekeeping system [34], while the SUF was shown to be an oxidative stress or Fe-limited inducible system [35]. Indeed, the transcriptomic analyses showed the significant induction of numerous Fe acquisition mechanisms, including an iron/manganese transporter, ferrichrome, hemin, ferrioxamine, ferric coprogen and ferric-rhodotorulic acid uptake systems, then biosynthesis and uptake of salmochelin, enterobactin, as well as hydroxamate siderophores, clearly indicating a state of Fe-limitation in oxidatively stressed cells. These data showed the activation of the various Fe-acquisition systems in *S*. Enteritidis cells during oxidative stress, further suggesting their critical importance in cellular adaptation to oxidative stress.

Hydroxyl free radical, a product of the Fenton reaction, represents the most damaging compound during H_2_O_2_ treatment [36]. This highly reactive oxygen species interacts with numerous biomolecules further causing multiple effects on the treated prokaryotic cells [37]. The comparative transcriptomics analyses showed multifaceted responses of oxidatively treated cells including induction of not only the OxyR (*katG*, *ahpCF*) and the SoxRS (*soxR*, *soxS*, *sodA*, *sodC*), two oxidative stress regulons, but also stress responses involved in DNA and protein repairs, osmoprotection, multidrug efflux, phosphate and magnesium starvation. During the oxidative stress response, this human pathogen induced a robust DNA repair system, which included base repair (*deoBCD*), single base mismatch repair (*dinIG*), early double-strand break repair (*yebG*), multiple double-strand breaks repair (*recAN*) and DNA replication (*polB*, *dinP*). Also, the *dps* gene which encodes a multifunctional Dps protein was greatly induced. Dps non-specifically binds DNA, forming a highly stable Dps-DNA nucleoprotein complex which further protects DNA from the deleterious effects of ROS [38]. Similarly to DNA, protein repair systems were induced. The *ibpAB* genes, encoding small heat-shock proteins IbpA and IbpB, showed the greatest induction among various protein repair systems. Induction of genes encoding molecular chaperones (*dnaKJ*, *clpB*) and proteases (*clpA*, *pgtE*) indicates a tendency of proteins to aggregate or to undergo a process of denaturation during oxidative stress. Multifaceted responses of oxidatively treated cells was also reflected by induction of key genes associated with osmotic stress (*osmYC*, *envFR*), manganese starvation (*sitABCD*, *pagCDOK*, *mgtBC*), phosphate starvation (*phoEH*, *yjbA*, *phnTSUR*), multidrug efflux (*mdlAB*, *yhcQP*) and outer membrane modification (*wcaEFH*, *yncCJ*, *yjbAEHJ*). Adaptation of *S*. Enteritidis to oxidative stress was not only confined to the induced transcriptome but it also included the repressed transcriptome. The significant repression of the major small (*rpsUTPLGSOCFHNJQBR*) and large (*rplUKSJNXVBCDAEPLWYM*) units of ribosomal encoding genes indicates that oxidatively stressed cells activated a stringent response. The main purpose of a stringent response is saving cellular energy and precursors required for ribosomal synthesis. If cells continue to synthesize ribosomes, the majority of these organelles would end up being inactive due to the disruption of central metabolic pathways (i.e., lack of amino acids) and this would be an unnecessary and costly expenditure of resources [37].

Besides the alterations of a large pool of genes encoding various central metabolic and stress response pathways, the comparative transcriptome analyses revealed a clear division, during the H_2_O_2_ treatment, in the gene expressions associated with the genes encoding for motility (planktonic) and the adhesion genes (sessile) forms of life. The exposure of *S*. Enteritidis to oxidative stress had an extensive and significant positive effect on the expression of adhesion encoding genes, including fimbrial *stbABCD*, *fimHYWDF*, *sthB*, SEN4247; curli *csgABF*, and pilin *safBCD*, SEN1978 (type IV pilin), *yibEHG* and *yncC* operons and genes. At the same time, it negatively affected expressions of the major flagellar operons *flgHECFIGJDBMN* and *fliJMRLNKFQPOIHEBG* clearly indicating a switch from a planktonic to a sessile (i.e., biofilm) life forms of oxidatively treated cells. Recently, it was shown that a transition from planktonic to sessile life form in *S*. Enteritidis is followed by repression of the major flagellar operons and induction of numerous fimbrial, curli, and pili operons [22]. This transcriptional change is in agreement with the “swim-or-stick” theory, which postulates that motility and biofilm development are mutually-exclusive processes [39]. Our transcriptomic data indicate that oxidatively stressed cells transition from a planktonic to a biofilm form of life. Another important feature of oxidatively stressed cells was an extensive induction of salmonellae-essential virulence genes located on SPI 1, SPI 2, and SPI 5. Induction of core virulence genes involved genes encoding not only T3SS apparatus of the SPI 2 and translocon of SPI 1 but also numerous effectors (*sifAB*, *sseAbjbBFIE*, *iagB*, *sinHR*, *sopE2*, *sicP*) essential for intracellular survival of this pathogen [40]. Fu and colleagues [18] studying the oxidative stress response of *S.* Typhimurium found that this organism repressed the synthesis of core virulence proteins encoded on SPI 1. Discrepancy between their and our results can be most likely explained by different experimental setups. While Fu and colleagues [18] exposed the cultures of *S.* Typhimurium to H_2_O_2_ at their mid-exponential growth phases (0.5 at OD_600_) and harvested them at their early stationary growth phase (0.9 at OD_600_). We started H_2_O_2_ treatment at the mid-exponential growth phase (0.4 at OD_600_) and harvested the cultures at their still exponential growth phase (~0.5 at OD_600_). Taken together, our experimental approach ensured that all transcriptional changes will be consequences of oxidative stress, while the approach employed by Fu et al., [18] reflects not only oxidative stress response but additional stress responses associated with the stationary growth phase of this organism [41,42].

## 5. Conclusions

This study employing a combination of incremental oxidative stress assays and the next generation sequencing approach revealed additional levels of the oxidative stress response complexity in non-typhoidal *Salmonella*. It was shown that the oxidative stress response is intricately linked with not only Fe-S cluster formation, iron homeostasis, DNA repair, and cell redox homeostasis but also with heat, osmotic, stringent, manganese and phosphate starvation stress responses as well as multidrug efflux and decreased passive influx of small hydrophilic molecules. In addition to these multifaceted cellular adaptation processes, it was shown that *S*. Enteritidis during oxidative stress simultaneously repressed the key motility encoding genes and massively induced the essential adhesion and salmonellae core-virulence encoding genes, crucial for the biofilm formation and host cell invasion. Based on these data, we hypothesize that the oxidative stress response of non-typhoidal *Salmonella* is closely associated with intracellular survival and that oxidative stress may act as a stimulus for eukaryotic cell invasion and formation of *Salmonella*-containing vacuoles. It has been already found that intracellular biofilm formation plays an important step in virulence of uropathogenic *Escherichia coli*, another facultative intracellular pathogen [43], suggesting that a similar approach can be explored by non-typhoidal *Salmonella*.

## Figures and Tables

**Figure 1 antioxidants-09-00849-f001:**
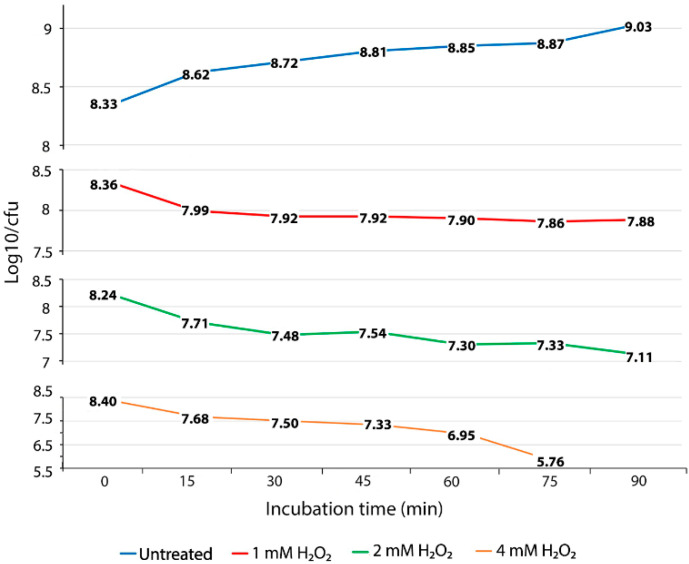
Oxidative stress killing assay. Mortality of *S.* Enteritidis, expressed by reduced log_10_/CFU for every 15 min of the H_2_O_2_ treatment. The data correspond to the mean value of three biological replications. Each measurement value was composed of nine replications (each biological replication was made up of three technical replications).

**Figure 2 antioxidants-09-00849-f002:**
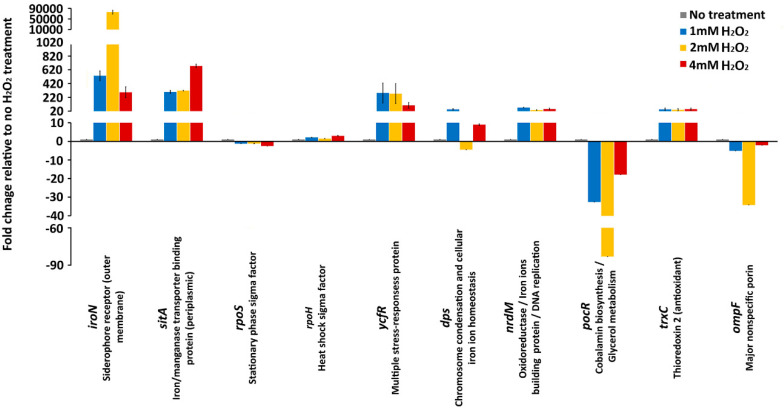
Gene expression. mRNA expression levels of *iroN*, *sitA* (iron receptor), *rpoS*, *rpoH* (alternative sigma factors), *ycfR*, *dps* (stress response), *nrdM*, *trxC* (antioxidant), *ompF* (nonselective uptake), and *pocR* (anabolism) during no H_2_O_2_ treatment, 1 mM, 2 mM, and 4 mM H_2_O_2_ treatments. Values on the y-axis are relative expression levels (fold change) normalized against the level in the no H_2_O_2_ treatment group. The data correspond to the mean values of three biological replications. Error bars correspond to the standard deviation.

**Figure 3 antioxidants-09-00849-f003:**
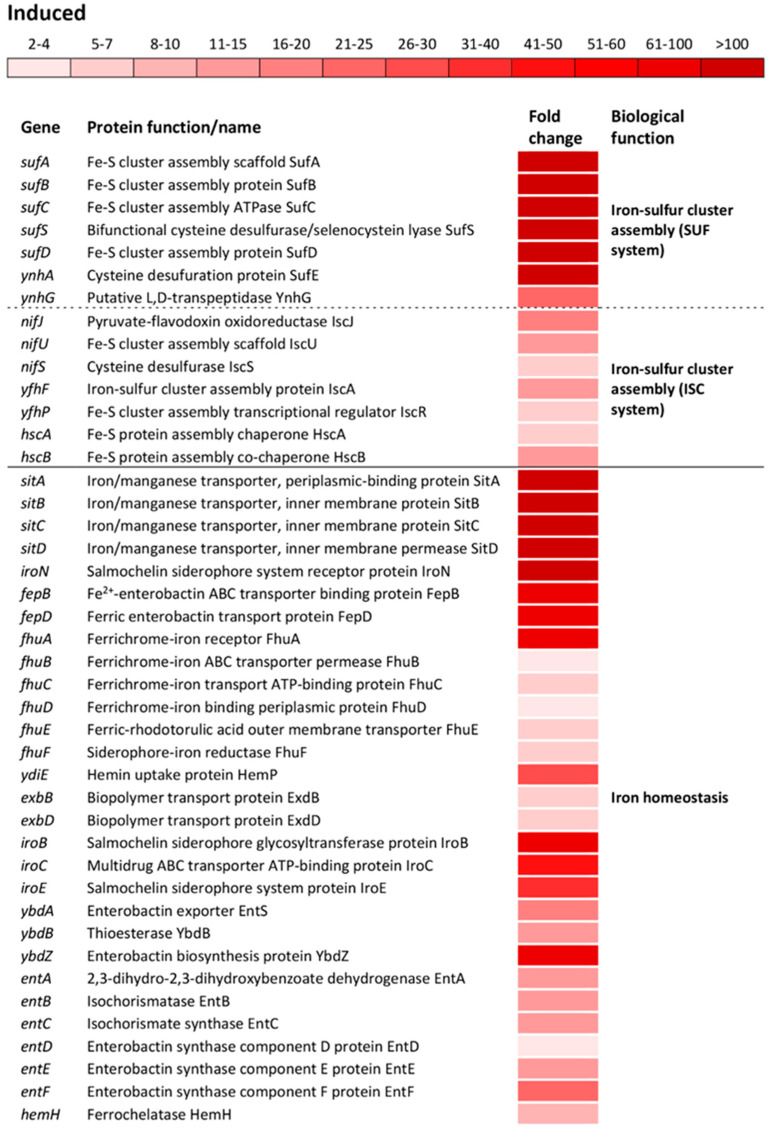
Differential expression of genes involved in iron-cluster assembly and iron ion homeostasis. Transcriptional profiling of the wild type *S*. Enteritidis ATCC 13076. Isolation of RNA was carried out during the exponential growth phase. The transcriptomic profiles were determined using a HiSeq 4000 PE100 Illumina platform. The mean fold-increase of three biological replications in each gene expression after exposure to 3 mM of H_2_O_2_ compared with controls (no H_2_O_2_ treatment and the same growth phase) is indicated by the color scale bar. The biological functions of the induced gene were determined using the Database for Annotation, Visualization and Integrated Discovery (DAVID) as well as the National Center for Biotechnology Information (NCBI). Both sulfur utilization factor (SUF) and iron-sulfur cluster (ISC) belong to the same biological function “Iron-sulfur cluster assembly”.

**Figure 4 antioxidants-09-00849-f004:**
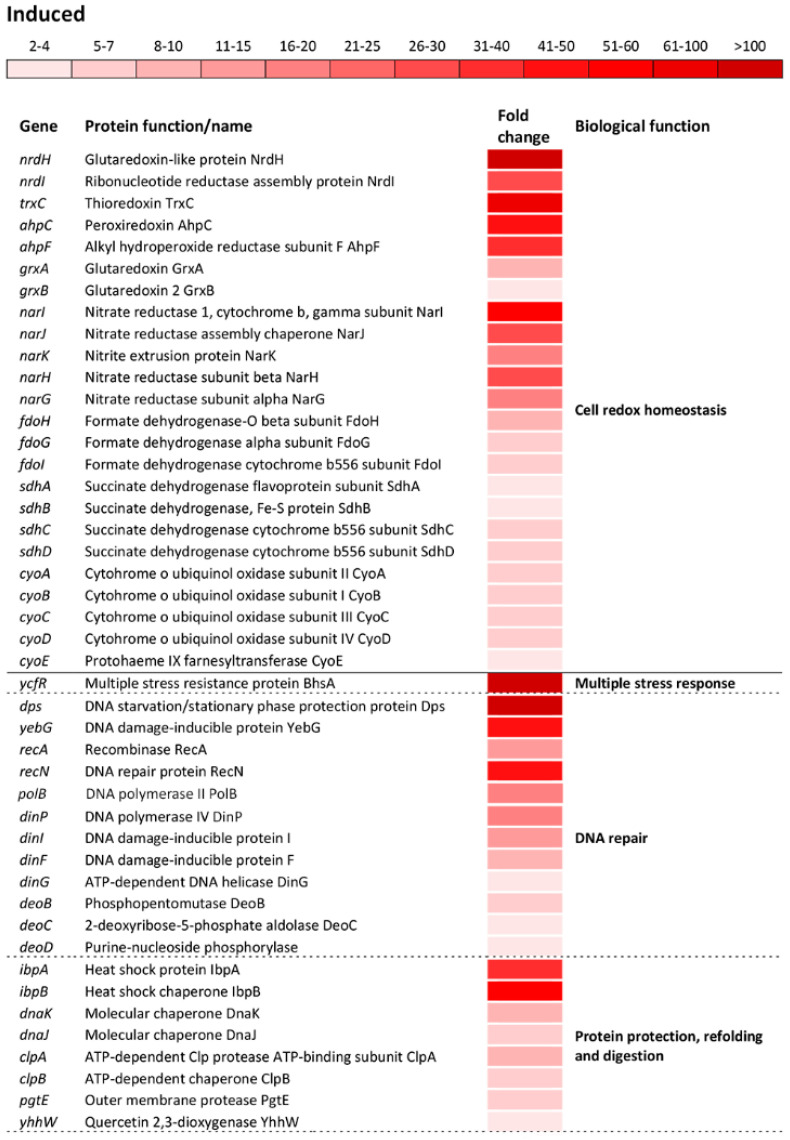
Differential expression of genes involved in cell redox homeostasis and various stress responses. The mean fold-increase in each gene expression after exposure to 3 mM of H_2_O_2_ compared with controls is indicated by the color scale bar. Note that a variety of biological functions including multiple stress response, DNA repair, protein protection, refolding and digestion, osmoprotectants, efflux, phosphate starvation, removal of reactive oxygen species, response to low Mg^2+^ environment and programed cell death belong to a more general biological function called “Stress response”.

**Figure 5 antioxidants-09-00849-f005:**
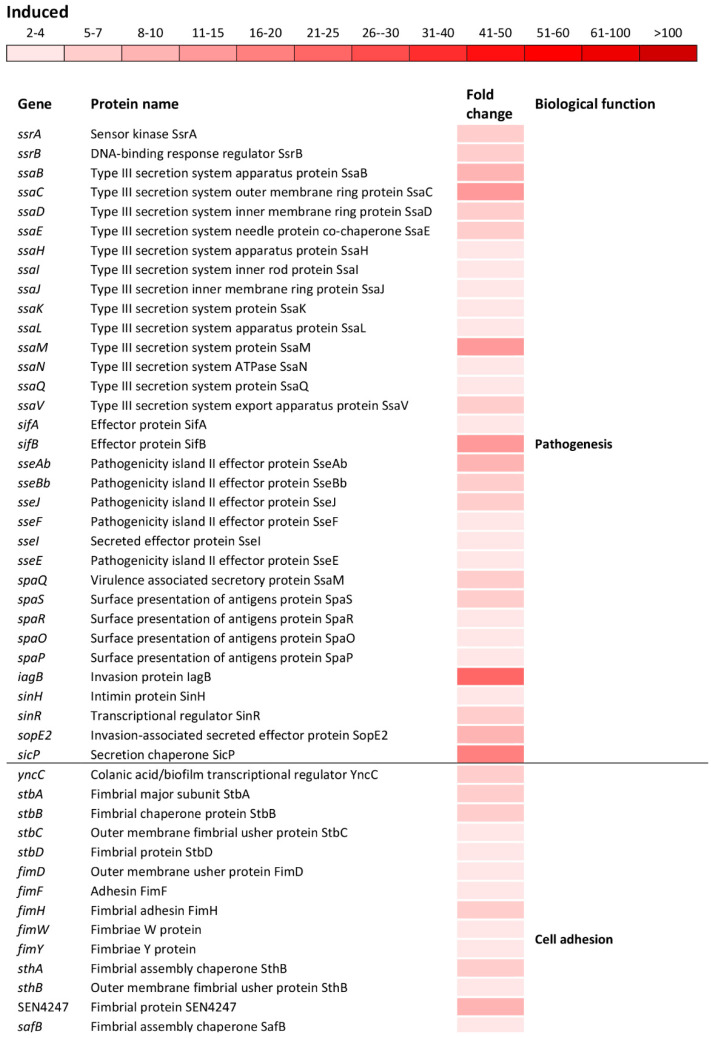
Differential expression of genes involved in pathogenesis and cell adhesion. The mean fold-increase in each gene expression after exposure to 3 mM of H_2_O_2_ compared with controls is indicated by the color scale bar.

**Figure 6 antioxidants-09-00849-f006:**
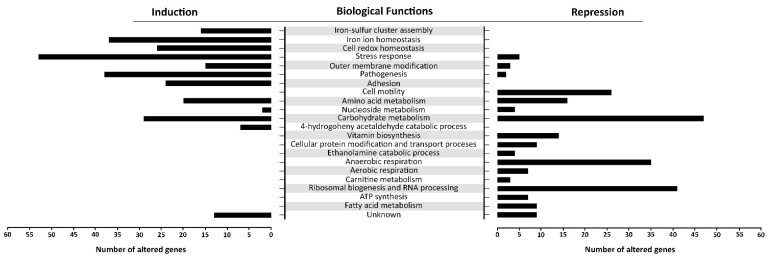
Gene ontology (GO) enrichment analysis. The figure portrays the most altered biological functions of *S*. Enteritidis during the H_2_O_2_ treatment. In the GO analysis were included operons/genes that showed the greatest alterations compared to their counterparts in the control sample. Each gene showed false discovery rate (FDR) *p* < 0.05 and high reproducibility across the biological replications.

**Figure 7 antioxidants-09-00849-f007:**
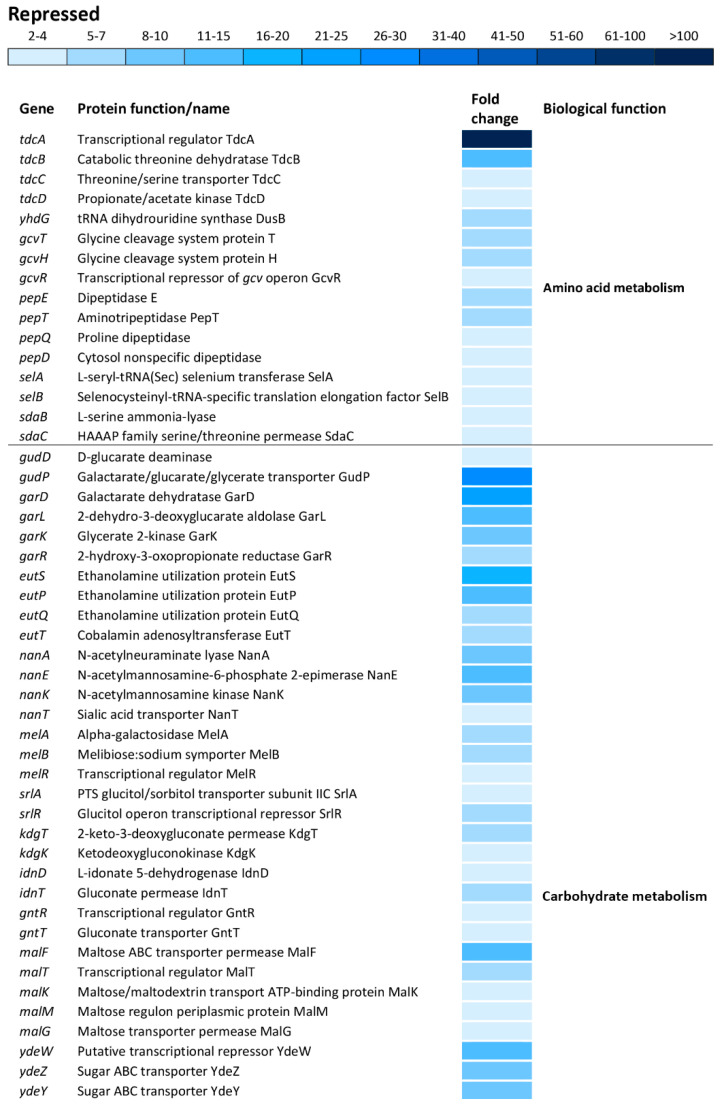
Differential expression of genes involved in amino acid and carbohydrate metabolisms. The mean fold-decrease in each gene expression after exposure to 3 mM of H_2_O_2_ compared with controls is indicated by the color scale bar.

**Figure 8 antioxidants-09-00849-f008:**
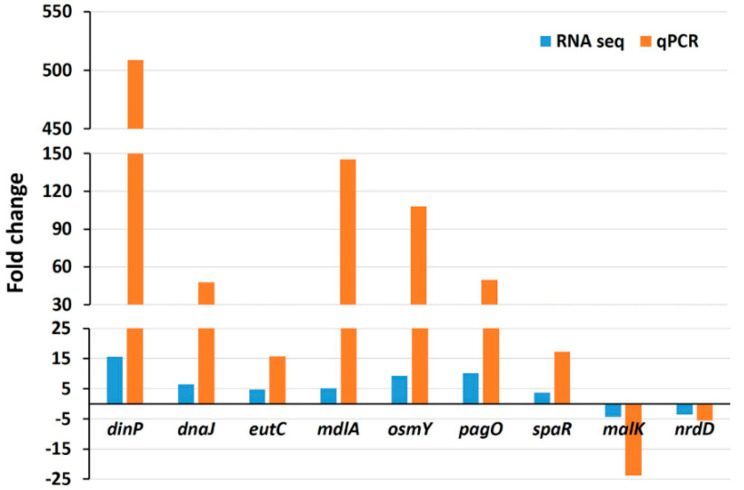
Validation of RNA-seq data by qRT-PCR analysis. Data represent fold changes in the expression of nine randomly selected genes between the wild type strain without treatment and the same strain treated with 3 mM H_2_O_2_. Genes differently expressed between the wild type strain with no treatment and the wild type with H_2_O_2_ treatment represent the mean value of three biological replications.

**Table 1 antioxidants-09-00849-t001:** Primers used in this study for the oxidative stress response gene expression assay and validation of RNA-seq data.

Gene Names	Primers for the Oxidative Stress Response Gene Expression Assay	
	Forward Sequence (5′–3′)	Reverse Sequence (5′–3′)
*rpoS*	CGA AAA AGC GTT GCT GGA CA	ATC GTC ATC TTG CGT GGT GT
*rpoH*	CCT TCG CCG TAC ACT GGA TT	GCT TTC GTG GTT GCG ACT TT
*iroN*	TCA TGG AAA ATG ACC CCG CA	ACC GCG TTC GAA GTA CTG TT
*sitA*	CGC CAA AAC AGG TGC GTA AA	ATC GGA AAC CGT ACT CTC GC
*dps*	AAA AGC GAC GGT TGA GTT GC	TAC GGA AGC CAT CCA GCA TC
*nrdH*	TGG TGA ACG TCG ATC TGG TG	CGG GTG CAG ACG GTT AAT CA
*trxC*	AGC GGT AAA GTC CGT TTC GT	GAA AGG CGC TTT AGG CAC TG
*ycfR*	CCG TTG AAG TTC AGG CAA CG	CCC ATC TCC TGC GCT TTT TG
*ompF*	CAG CGT ACA GCA ACA GCA AG	TCA GCA TAT ACG GCA GCC AG
*pocR*	GTG GGT AAA CCG CCA GAG AA	AGG TCT GGC GGA AGA CTT TG
**Primers for Validation of the RNA-Seq Data**		
*dinP*	TTA CGG GCG GTG GTA ATT AAG	AAG GCT GCG TAA ATC GGT AG
*osmY*	AAC TCT GCT GGC CGT AAT G	AGG GTG ACG ACT TTC TGA TTG
*dnaJ*	TGG ATC TCA CCC TGG AAG AA	GCC CAT GAC CGT GAC ATT TA
*mdlA*	CTC CAG TTG CTG ATA GCG ATA C	GCC AGC GTA CAT GAG GAT ATT
*eutC*	CTC AAA GAA GTG CCG GAA GA	GGC AGG ATC TCT TCA TAG TTG G
*spaR*	GAA AGA GAG TCG CGG TAC ATT	CCT TCA TTA CCG CCG CTA TTA
*nrdD*	CTA TCT GCC CGT TGC CTA AAT	GCT GGA AGA TGT CTG GGA TTA C
*malK*	CGA GCG TAC AGC TAC GAA AT	AGA GCG CAT AAG ACT GGA ATA C
*pagO*	CCC GAG ATA CAG GGT AGC TAA TA	CAG ATC GCG GGC TTA ACT ATC

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
