# Peer review of "Insights into the Oxidative Stress Response of *Salmonella enterica* serovar Enteritidis Revealed by the Next Generation Sequencing Approach"

_antioxidants, 2020, doi:10.3390/antiox9090849_

Round 1

Reviewer 1 Report

The manuscript of Liu et al. used different modern molecular tools to study the transcriptional response of Salmonella enterica serovar Enteritidis to H2O2 exposure. The study produced a lot of data, which are interesting, however, I have some concern about this work:

  • L25-L27: do you have some phenotypic proof of this statement; just transcriptional data are not enough.
  • L27-L28: although a word “indicate” is used in this sentence, in my opinion the abstract should presents only very solid data. The statement in this sentence still needs to be confirmed by experimental data, therefore in my opinion this statement is not suitable for abstract or should be a least rewritten in a way that this needs further study.
  • L67: H2O: H2O2
  • L95: some explanation should be added why these genes were targeted.
  • Why you used in transcriptome analysis treatment of cells with 3 mM H2O2? These data cannot be compared to other data produced in this manuscript, since in other experiments you used 1, 2 or 4 mM H2O2.
  • Figures 1A and 1B: both figures present the same data but in different ways what is in my opinion not necessary. In my opinion Fig. 1B is enough, but here the statistics should be added, and y-axis should be unified.
  • The authors are writing in the first part of results that at some points the applied concentrations of H2O2 are bactericidal. How did the authors take into account this fact when interpreting the transcriptional data?
  • Section Results and Discussion are separated causing that quite some data on gene upregulation/downregulation are repeated. Maybe the authors can consider to combine both sections, it would be easier to follow the text for readers.
  • L461: do you have some phenotypic proof of this statement?
  • L494-L501: this is hypothesis which needs much of further experiments and therefore it needs to be written in much more careful way.

I am not sure if the subject of this manuscript fits into the journal Antioxidants.

Author Response

Reviewer 1

The manuscript of Liu et al. used different modern molecular tools to study the transcriptional response of Salmonella enterica serovar Enteritidis to H2O2 exposure. The study produced a lot of data, which are interesting, however, I have some concern about this work:

Response # 1. Thank you.

L25-L27: do you have some phenotypic proof of this statement; just transcriptional data are not enough.

Response # 2. The first part (underlined) of this statement “Importantly, this study revealed that under oxidative stress, S. enterica Enteritidis carry out a transition from a planktonic to a biofilm form of life followed by extensive induction of salmonellae-essential virulence genes critical for intracellular survival.” has a broad meaning that could not be entirely supported by the data from this study. While the second part of the same statement is based on the induction of virulence genes which is well documented in this study.

To avoid any possible misleading of the reader, this statement has been re-written. Now, it reads “Importantly, this study revealed that oxidatively stressed S. enterica Enteritidis cells simultaneously repressed key motility encoding genes and induced a wide range of adhesin and salmonellae-essential virulence encoding genes, critical for the biofilm formation and intracellular survival, respectively.” This new statement is entirely based on the data presented in this manuscript and should leave no doubt in the reader’s mind.

L27-L28: although a word “indicate” is used in this sentence, in my opinion the abstract should presents only very solid data. The statement in this sentence still needs to be confirmed by experimental data, therefore in my opinion this statement is not suitable for abstract or should be a least rewritten in a way that this needs further study.

Response # 3. Yes, it all depends on the reader’s perception of the word “indicate”. This statement is in the domain of the hypothesis mentioned in the conclusion. To clarify this point, we re-wrote this sentence. Now, it reads: “This finding indicates a potential intrinsic link between oxidative stress and pathogenicity in non-typhoidal Salmonella that needs to be empirically evaluated.”  

L67: H2O: H2O2

Response # 4. It says “H2O2”. We have checked the entire text. Each formula for hydrogen peroxide has been correctly written.

L95: some explanation should be added why these genes were targeted.

Response # 5. Agreed. The explanation has been provided. Now, this paragraph reads: “Primers were designed to target expression of the following genes: rpoS, rpoH, iroN, sitA, dps, nrdM, trxC, ycfR, ompF and pocR (Table 1). These ten genes were selected based on their distinct gene ontologies (GO) associated with the oxidative stress response. Selected the GO include, iron homeostasis, alternative sigma factors, multiple stress response, DNA protection, antioxidants, DNA replication, uptake of small solutes and anabolic processes.”

Why you used in transcriptome analysis treatment of cells with 3 mM H2O2? These data cannot be compared to other data produced in this manuscript, since in other experiments you used 1, 2 or 4 mM H2O2.

Response # 6. We did not have the intention to compare the global transcriptomic data (3 mM) with the incremental oxidative stress gene expression assay (1, 2 and 4 mM) data. With the incremental oxidative assay, we aimed to determine the gene expression dynamics of the selected genes during an incremental increase of H2O2 concentrations, whereas in the global transcriptomic approach we aimed to reveal the cellular response to the oxidative treatment using a single H2O2 concentration.

Regarding the concentration of 3 mM of H2O2 for the global transcriptomic study, it was selected based on the oxidative stress killing assay results and the literature. The authors usually use a range of 2 to 5 mM of H2O2 concentrations in their studies. Using this particular Salmonella strain, we found that 4 mM seemed to be too great concentration, while 2 mM showed a lot of similarity to 1 mM that imposed the bacteriostatic effect on this Salmonella strain. For these reasons, we selected a concentration of 3 mM, which represents a concentration that will impose a significant oxidative stress, but yet give a time to cells to adapt to the oxidative agent.       

Figures 1A and 1B: both figures present the same data but in different ways what is in my opinion not necessary. In my opinion Fig. 1B is enough, but here the statistics should be added, and y-axis should be unified.

Response # 7. Yes, these two graphs present the same set of data. A reason to include both of them was a fact that in Fig. 1A, the reader could see the fluctuations of the data (standard deviations), whereas in Fig. 1B it was difficult to incorporate. We agree that Fig. 1B is a major part of this figure. For that reason, Fig 1B is left in the manuscript, whereas Fig. 1A is presented as a supplementary figure. We still believe that a certain number of readers may appreciate to see it.   

The authors are writing in the first part of results that at some points the applied concentrations of H2O2 are bactericidal. How did the authors take into account this fact when interpreting the transcriptional data?

Response # 8. The main aim of this experiment was to determine the expression dynamics of the selected genes during the incremental increase of H2O2 concentration. What can be observed is that certain genes (trxC and nrdM) keep the same level of their gene expressions regardless of H2O2 concentration, whereas some genes decrees (ycfR) or increase (sitA) their expressions regarding the increase of H2O2 concentration. We cannot point this phenomenon to the oxidative susceptibility phenotype of Salmonella but rather likely to the role of these genes during the oxidative stress adaptation. As we mentioned in the manuscript, not only the time of expression but also gene expression dynamics play a role in the oxidative stress response. In other words, gene expression dynamics of oxidative stress response associated genes indicates another level of oxidative stress response complexity.        

Section Results and Discussion are separated causing that quite some data on gene upregulation/downregulation are repeated. Maybe the authors can consider to combine both sections, it would be easier to follow the text for readers.

Response # 9. Originally, we intended to combine “Results” and “Discussion” but after a careful reading of the journal’s Aims “Our aim is to encourage scientists to publish their experimental and theoretical results in as much detail as possible. Therefore, there is no restriction on the length of papers.” and further thinking about the impact of this study, we have changed our mind. As we employed a powerful approach to study the oxidative stress response of this human pathogen, we aimed to exploit the wealth of data and subsequently discrete the multifaceted nature of the oxidative stress adaptation. For these reasons we separated “Results” from “Discussion”. Of course, we ensured that there is no repetition between the “Results” and “Discussion”. We used the names of the genes in the “Discussion” only to reinforce certain points. For example, to emphasize what type of virulence genes were induced during the oxidative treatment, we presented to the reader names of genes that not only encode for the type 3 secretion system (base, needle) but also genes that encode effector proteins which are directly involved in Salmonella invasion. We believe that this type of study requires separation “Results” from “Discussion” so that the reader can in a stepwise approach acquire information from this manuscript.         

L461: do you have some phenotypic proof of this statement?

Response # 10. This sentence has been clarified. Now it reads: “Besides the alterations of a large pool of genes encoding various central metabolic and stress response pathways, the comparative transcriptome analyses revealed a clear division, during the H2O2 treatment, in the gene expressions associated with the genes encoding for motility (planktonic) and the adhesion genes (sessile) forms of life.”

With this modification, it becomes clear that the authors of this manuscript point at the differences in the gene expressions of the genes encoding adhesins and flagella during the oxidative treatment. 

L494-L501: this is hypothesis which needs much of further experiments and therefore it needs to be written in much more careful way.

Response # 11. Agreed. This paragraph is connected to the reviewer’s second comment. In both “Abstract” and “Conclusion”, we interpreted the transcriptomic changes of the H2O2 treated Salmonella cells as “transition from a planktonic to a biofilm form of life”. As this statement is purely based on the global transcriptomic data, it may mislead the reader since there are no biofilm related studies to confirm it. Therefore this point in the conclusion has been modified and now it reads: “In addition to these multifaceted cellular adaptation processes, it was shown that S. Enteritidis during oxidative stress simultaneously repressed the key motility encoding genes and massively induced the essential adhesion and salmonellae core-virulence encoding genes, crucial for the biofilm formation and host cell invasion. Based on these data, we hypothesize that the oxidative stress response of non-typhoidal Salmonella is closely associated with intracellular survival and that oxidative stress may act as a stimulus for eukaryotic cell invasion and formation of Salmonella-containing vacuoles. It has been already found that intracellular biofilm formation plays an important step in virulence of uropathogenic Escherichia coli, another facultative intracellular pathogen [43], suggesting that a similar approach can be explored by non-typhoidal Salmonella.”

Regarding the need for further experimentation to prove the hypothesis. Any hypothesis needs to be proved, that’s the essence of a hypothesis. The authors based on their data and observations coming up with different hypotheses, which subsequently may be proved. In this case, based on the global transcriptomic data, we proposed a hypothesis, which can be or cannot be proved in a new study by our or some other research group.       

I am not sure if the subject of this manuscript fits into the journal Antioxidants.

Response # 12. According to the scope of this journal this manuscript can be directly related to one area, “antioxidant metabolism in biological systems.” The entire study aims to determine changes in the metabolism of Salmonella enterica during oxidative stress. As this study showed, the antioxidants are only one facet of the multifaceted response of this human pathogen to the oxidative treatment.  

Reviewer 2 Report

The manuscript is well done and overall deserves
but I have some points to clarify

-Salmonella must always be written in the same way and in italics

-Why did you choose a 3M H2O2 concentration in the transcriptomics experiments that was not tested in your previous experiments?

-In the discussion, the physiological significance of the increase and repression of genes especially not involved in oxidation should be better explained

Author Response

Reviewer 2

The manuscript is well done and overall deserves
but I have some points to clarify.

Response # 13. Thank you.

Salmonella must always be written in the same way and in italics

Response # 14. Agreed. As the Salmonella nomenclature can be confusing, we followed the practice of the Centers for Disease Control and Prevention, USA and the instructions written by Brenner et al., Salmonella nomenclature. J Clin Microbiol. 2000, v.38, p.2465-2467. In other words, the full name of this Salmonella serovar was written at the first citation. After that, the name was written with the genus followed by the serovar name, for instance, Salmonella Enteritidis or S. Enteritidis. The entire text has been checked and modified accordingly.  

Why did you choose a 3M H2O2 concentration in the transcriptomics experiments that was not tested in your previous experiments?

Response # 15. Please see our detailed response # 6 to this question.

In the discussion, the physiological significance of the increase and repression of genes especially not involved in oxidation should be better explained.

Response # 16. We think that we have provided an extensive explanation of the altered genes in the “Results” section, while in the “Discussion” section we attempted to point out at the novelties that this study revealed. Because of the length of this manuscript, we did not want to focus on the facts already known that will subsequently extend the manuscript and potentially dilute out the reader’s focus.  

Round 2

Reviewer 1 Report

The manuscript has been corrected accordingly.